

# Scientific publications in nursing journals from Mainland China, Taiwan, and Hong Kong: a 10-year survey of the literature

Di Zhang[1],[*], Xiaming Wang[2],[*], Xueru Yuan[1],[*], Li Yang[1], Yu Xue[1] and Qian Xie[1]

[1] Department of Anesthesiology, Navy General Hospital, Beijing, China
[2] Department of Otorhinolaryngology, Navy General Hospital, Beijing, China
[*] These authors contributed equally to this work.

## ABSTRACT

**Background:** China has witnessed remarkable progress in scientific performance in recent years. However, the quantity and quality of nursing publications from three major regions (Mainland China, Taiwan, and Hong Kong) have not been reported. This study aimed to investigate the characteristics of scientific research productivity from Mainland China, Taiwan, and Hong Kong in the field of nursing. **Methods:** Articles published in the 110 nursing journals originating from Mainland China, Taiwan, and Hong Kong between 2005 and 2014 were retrieved from the Web of Science. The total number of articles published, the impact factor, and the citation count were analyzed. **Results:** There were 2,439 publications between 2005 and 2014 from China, including 438 from Mainland China, 1,506 from Taiwan, and 495 from Hong Kong. There was a significant increase in publications for these three regions (p < 0.05), especially for Mainland China, with a 59.50-fold increase experienced. From 2011, the number of publications from Mainland China exceeded that from Hong Kong. Taiwan had the highest total journal impact factor (2,142.81), followed by Hong Kong (720.39) and Mainland China (583.94). The mean journal impact factor from Hong Kong (1.46) was higher than that from Taiwan (1.42) and Mainland China (1.33). Taiwan had the highest total citation count (8,392), followed by Hong Kong (3,785) and Mainland China (1,493). The mean citation count from Hong Kong (7.65) was higher than that from Taiwan (5.57) and Mainland China (3.41). The Journal of Clinical Nursing was the most popular journal in the three regions. **Discussion:** Chinese contributions to the field of nursing have significantly increased in the past ten years, particularly from Mainland China. Taiwan is the most productive region in China. Hong Kong had the highest-quality research output, according to mean journal impact factor and mean citation count.

# INTRODUCTION

China is a rapidly progressing developing country, with a population of over 1.3 billion people. In this context of rapid economic development, China has played an increasing

Corresponding authors
Yu Xue, xueynavy@163.com
Qian Xie, xieqiannavy@163.com

role in medicine and science in the recent decades (*Tong, Wang & Jiang, 2013*; *Cyranoski, 2004*; *Ding, Jia & Liu, 2015*; *Liang et al., 2015*; *Luo et al., 2015*). A similar trend has also been observed with China's contribution to nursing research (*Li et al., 2009*; *Li, 2014*; *Kalisch & Kalisch, 2009*). Moreover, there are more than 2 million registered nurses in China (*Gao, Chan & Cheng, 2012*). China may become an important hub in worldwide nursing research, given the emerging status and large community of practitioners found in the country.

There are three major Chinese-speaking regions: Mainland China, Taiwan, and Hong Kong. Although people in these regions share the same ethnic origin of Han Chinese, they have distinct political regimes, economic status, and health care systems (*Xu, Xu & Zhang, 2000*; *Lin et al., 2014*). These characteristics may lead to different scientific research productivity outputs in these regions. Publication, as a central part of scientific research, is an important indicator of research productivity. Recently, it has been widely used to compare the scientific productivity of these three regions in many medical fields (*Gao, Liao & Li, 2008*; *Li et al., 2010*; *Zheng et al., 2011*; *Cheng & Zhang, 2010*). However, as far as we are aware, the quantity and quality of nursing research production in these Chinese-speaking regions have not yet been reported. Therefore, the present study aimed to investigate the characteristics of scientific publications in the field of nursing from the major Chinese-speaking regions, specifically Mainland China, Hong Kong, and Taiwan, over a ten-year period.

# MATERIALS AND METHODS

## Search strategy

The present study was designed based on methodology of previous similar articles (*Gao, Liao & Li, 2008*; *Li et al., 2010*; *Zheng et al., 2011*; *Cheng & Zhang, 2010*). In June 2015, the Web of Science Core Collection was used to carry out a computerized search in order to identify nursing publications. There are 110 subspecialty nursing journals listed under the "Nursing" category of the 2014 Journal Citation Report (JCR). To include all articles published in the 110 nursing journals, the titles of the journals were placed in the search window using the "OR" operator. The study period was limited to 2005–2014. The refine panel of Web of Science was used to identify articles. The filter of "Country/Region" was used to firstly sort the articles published by different regions. We then selected "Peoples R China" or "Taiwan." Thus, articles originating from China between 2005 and 2014 in these journals were identified. The filter of "Document types" was then used to further refine the results. Only original articles and reviews were included. Letters, editorial material, and corrections were excluded. The search output was exported to Microsoft Office Excel for further analysis. The "Reprint Address" for each article was considered as the source region (*Zheng et al., 2011*). Articles with are print address located in Mainland China, Taiwan, and Hong Kong were selected accordingly.

## Data extraction

Two reviewers independently conducted the study selection and data extraction. The title and the abstract of potentially eligible articles were reviewed. Articles unrelated to nursing

topics were further excluded. Disagreements were resolved by discussion. The number of publications was used to evaluate the quantity of research output. The quality of research productivity was assessed through two indicators: journal impact factor and citation count of articles. The impact factor was obtained from the 2014 JCR. A region's total impact factor of published papers was calculated by multiplying each journal's impact factor by the number of articles contained within the publication. From articles originating in Mainland China, Taiwan, and Hong Kong, the following information was collected: the total number of articles, the total and mean journal impact factor, the total number of citations and mean citation count per article, articles published in the top ten high-impact journals, and the five most popular journals in the three regions. Journal popularity in a region was defined according to the number of articles published by authors from this region: the more the articles, the more popular the journal.

### Statistical analysis

Descriptive statistics (e.g. totals and means) are mainly used in this study. Regression analysis was used to determine relationships between the quantities of articles published, and time trends from 2005 to 2014. Data were analyzed with SPSS version 19.0 (SPSS Inc., Chicago, IL, USA) and a significance threshold of 0.05.

## RESULTS

### Total number of articles

From the 110 nursing journals assessed between 2005 and 2014, a total number of 2,439 publications originated from China. The annual total numbers of articles from China increased significantly from 2005 to 2014 (from 105 to 364; $p = 0.000$), with a 3.47-fold increase experienced. Taiwan published the largest number of articles in this time (1,506/2,439; 61.75%), followed by Hong Kong (495/2,439; 20.30%) and Mainland China (438/2,439; 17.96%). Figure 1 shows the published articles from Mainland China, Hong Kong, and Taiwan. Mainland China had the most rapid increase in the number of annual publications (from 2 to 119; $p = 0.000$), which is a 59.50-fold increase, followed by Taiwan (from 68 to 195; $p = 0.001$) and Hong Kong (from 35 to 50; $p = 0.017$). From 2011, the number of papers published from Mainland China exceeded those from Hong Kong; however, Mainland China had fewer publications than Taiwan.

### Journal impact factor

According to the 2014 JCR (Table 1), the total journal impact factor of Taiwan's published papers (2,142.81) was higher than that of Hong Kong (720.39) and Mainland China (583.94). However, Hong Kong had the highest mean journal impact factor, at 1.46, followed by Taiwan (1.42), and Mainland China (1.33) (Table 1). The impact factor of the journals publishing articles from Mainland China was between 0.220 and 2.901; the range for Taiwan was from 0.000 to 2.901; and Hong Kong was between 0.439 and 2.901.

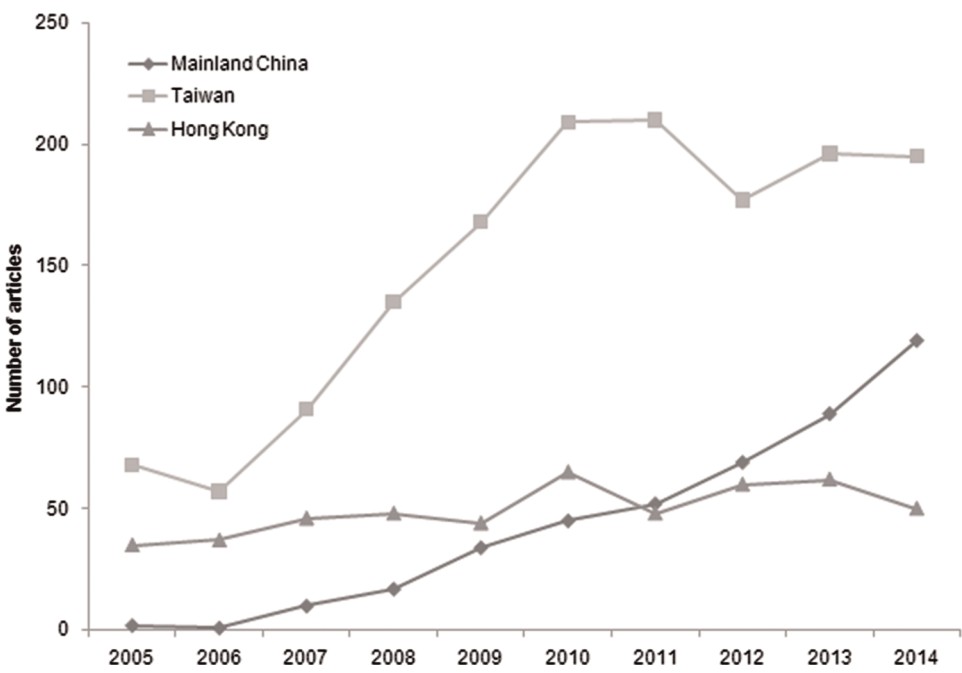

**Figure 1** The number of articles from Mainland China, Taiwan, and Hong Kong in subspecialty nursing journals of 2014 Journal Citation Reports.

**Table 1** Total and mean impact factor of articles from Mainland China, Taiwan, and Hong Kong.

|  | Total impact factor | | | Mean impact factor | | |
| --- | --- | --- | --- | --- | --- | --- |
| Year | Mainland China | Taiwan | Hong Kong | Mainland China | Taiwan | Hong Kong |
| 2005 | 3.71 | 104.80 | 50.17 | 1.85 | 1.54 | 1.43 |
| 2006 | 1.26 | 98.22 | 57.64 | 1.26 | 1.72 | 1.56 |
| 2007 | 13.52 | 136.66 | 71.06 | 1.85 | 1.50 | 1.54 |
| 2008 | 21.53 | 200.38 | 69.33 | 1.27 | 1.48 | 1.44 |
| 2009 | 46.64 | 238.29 | 60.56 | 1.37 | 1.42 | 1.38 |
| 2010 | 65.87 | 307.73 | 101.24 | 1.46 | 1.47 | 1.56 |
| 2011 | 74.07 | 296.70 | 70.35 | 1.42 | 1.41 | 1.47 |
| 2012 | 93.76 | 243.55 | 83.21 | 1.36 | 1.38 | 1.39 |
| 2013 | 112.08 | 260.92 | 89.25 | 1.26 | 1.33 | 1.44 |
| 2014 | 151.52 | 255.56 | 67.59 | 1.27 | 1.31 | 1.35 |
| Total | 583.94 | 2142.81 | 720.39 | 1.33 | 1.42 | 1.46 |

## Citation count

Taiwan had a higher citation count of publications between 2005 and 2014 than Hong Kong (3,785) and Mainland China (1,493) (Table 2). However, Hong Kong had the highest mean citation count per article (7.65), followed by Taiwan (5.57), and Mainland China (3.41) (Table 2).

**Table 2** Total and mean citation of articles from Mainland China, Taiwan, and Hong Kong.

| | Total citation | | | Mean citation | | |
|---|---|---|---|---|---|---|
| Year | Mainland China | Taiwan | Hong Kong | Mainland China | Taiwan | Hong Kong |
| 2005 | 28 | 950 | 607 | 14.00 | 13.97 | 17.34 |
| 2006 | 28 | 732 | 436 | 28.00 | 12.84 | 11.78 |
| 2007 | 174 | 1040 | 609 | 17.40 | 11.43 | 13.24 |
| 2008 | 151 | 1417 | 577 | 8.88 | 10.50 | 12.02 |
| 2009 | 256 | 1159 | 505 | 7.53 | 6.90 | 11.48 |
| 2010 | 201 | 1261 | 490 | 4.47 | 6.03 | 7.54 |
| 2011 | 195 | 995 | 227 | 3.75 | 4.74 | 4.73 |
| 2012 | 274 | 481 | 185 | 3.97 | 2.72 | 3.08 |
| 2013 | 127 | 255 | 133 | 1.43 | 1.30 | 2.15 |
| 2014 | 59 | 102 | 16 | 0.50 | 0.52 | 0.32 |
| Total | 1493 | 8392 | 3785 | 3.41 | 5.57 | 7.65 |

## High-impact worldwide nursing journals

In the 2014 JCR, the journal impact factor of ten high-impact worldwide nursing journals in which Chinese authors published their work was greater than 1.7 (Table 3). Mainland China, Taiwan, and Hong Kong had a total of 627 publications appear in these ten high-impact journals between 2005 and 2014. Among these journals, *Journal of Advanced Nursing* had the highest number of articles (256), followed by *International Journal of Nursing Studies* (191), and *Cancer Nursing* (100). Taiwan published the largest number of papers (392) in the ten high-impact journals, followed by Hong Kong (149) and Mainland China (86).

## Popular nursing journals

The five most popular journals in the field of nursing for China are shown in Table 4. *Journal of Clinical Nursing* was the most popular journal in all three regions, with 74 articles originating from Mainland China, 392 articles from Taiwan, and 146 articles from Hong Kong. *Journal of Clinical Nursing*, *Journal of Advanced Nursing*, *International Journal of Nursing Studies*, and *Cancer Nursing* appeared in all top five journals across all three regions. *Nurse Education Today* was found in the top five journal list in two regions. *Journal of Nursing Research* appeared in the top five journal list in one region.

## DISCUSSION

To our knowledge, this is the first study to analyze the quantity and quality of articles in major nursing journals that have been published from three major Chinese regions, namely Mainland China, Taiwan, and Hong Kong, during a ten-year period. This survey provides a general picture of Chinese research output in the field of nursing. This study revealed that China became increasingly productive from 2005 to 2014 in major nursing journals, particularly Mainland China. Scientific production in the field of nursing from Mainland China exceeded the output from Hong Kong

**Table 3 Articles published in the 10 top-ranking worldwide journals from Mainland China, Taiwan, and Hong Kong.**

| Rank | Journal | Impact factor | Mainland China | Taiwan | Hong Kong | Total |
|------|---------|---------------|----------------|--------|-----------|-------|
| 1 | International Journal of Nursing Studies | 2.901 | 22 | 132 | 37 | 191 |
| 2 | Oncology Nursing Forum | 2.788 | 1 | 8 | 2 | 11 |
| 3 | Worldviews on Evidence-Based Nursing | 2.381 | 0 | 9 | 1 | 10 |
| 4 | American Journal of Critical Care | 2.115 | 1 | 10 | 3 | 14 |
| 5 | Journal of Cardiovascular Nursing | 2.053 | 6 | 12 | 3 | 21 |
| 6 | Journal of Human Lactation | 1.985 | 4 | 2 | 2 | 8 |
| 7 | Cancer Nursing | 1.966 | 17 | 64 | 19 | 100 |
| 8 | International Journal of Mental Health Nursing | 1.950 | 5 | 2 | 3 | 10 |
| 9 | European Journal of Cardiovascular Nursing | 1.876 | 0 | 5 | 1 | 6 |
| 10 | Journal of Advanced Nursing | 1.741 | 30 | 148 | 78 | 256 |
| Total | | | 86 | 392 | 149 | 627 |

**Table 4 The five most popular nursing journals in Mainland China, Taiwan, and Hong Kong.**

| Rank | Mainland China | N | Taiwan | N | Hong Kong | N |
|------|----------------|---|--------|---|-----------|---|
| 1 | Journal of Clinical Nursing | 74 | Journal of Clinical Nursing | 392 | Journal of Clinical Nursing | 146 |
| 2 | Journal of Advanced Nursing | 30 | Journal of Nursing Research | 167 | Journal of Advanced Nursing | 78 |
| 3 | International Journal of Nursing Studies | 22 | Journal of Advanced Nursing | 148 | International Journal of Nursing Studies | 37 |
| 4 | Nurse Education Today | 18 | International Journal of Nursing Studies | 132 | Nurse Education Today | 29 |
| 5 | Cancer Nursing | 17 | Cancer Nursing | 64 | Cancer Nursing | 19 |
| Total | | 178 | | 952 | | 328 |

since 2011. Taiwan is the most prolific region in China. However, Hong Kong published the highest-quality nursing research, according to the analysis of mean journal impact factor, and mean citation count per paper.

China's increasing contribution to scientific research has been demonstrated in many biomedical fields (*Gao, Liao & Li, 2008*; *Li et al., 2010*; *Zheng et al., 2011*; *Cheng & Zhang, 2010*). This also holds true for the field of nursing, according to the present findings. Although Mainland China has lagged behind Taiwan and Hong Kong in terms of scientific productivity for many years, since 2011, scientific research production in the field of nursing from Mainland China has exceeded production from Hong Kong (*Gao, Liao & Li, 2008*; *Li et al., 2010*; *Zheng et al., 2011*; *Cheng & Zhang, 2010*). Furthermore, the most rapid increase in the number of nursing publications was seen in articles emanating from Mainland China. There are several possible reasons for these findings. First, a central reason may be the rapid development of China's economy, which has induced increased funding to the field of nursing (*Kalisch & Kalisch, 2009*; *Gao, Chan & Cheng, 2012*). Second, Chinese nursing researchers have more experience than before, and the ability of nurses in conducting research has been improved (*Li, 2014*; *Kalisch & Kalisch, 2009*; *Gao, Chan & Cheng, 2012*). Third, high-level research institutions are increasingly being established in China, and the education level of nursing is rapidly

rising (*Kalisch & Kalisch, 2009*; *Gao, Chan & Cheng, 2012*). The improvement of research infrastructure and the quality of researchers may have prompted the high-level research output observed in this study. Fourth, language has previously constituted an important barrier for most Chinese nursing researchers. However, with increasing international knowledge-exchange, more authors from China have recently published their work in international journals, thanks to increased English proficiency, and developments in the experience of medical research (*Cheng, 2012*; *Jia et al., 2015*; *Tong, Wang & Jiang, 2013*). For these reasons, China will be able to contribute more to scientific publication in the field of nursing.

Although China has increasing contributions to the field of nursing, several current problems facing the discipline should be recognized. First, shortages of nurses are a global problem, and China faces a severe shortage (*Zeng, 2009*). Nurses in China may lack sufficient time to perform high-quality research. Second, although nursing education in China has developed rapidly, the overall level of nursing education is still relatively low, requires improvement (*You et al., 2015*). Having said this, these problems are being addressed in China (*Zeng, 2009*; *You et al., 2015*). It is hoped that, through medical reforms, the status of nursing research in China will be improved in the future (*Chen, 2009*; *Liu, 2009*).

In this study, journal impact factor and citation count were selected as measures, as they have been widely used to evaluate the quality of articles, as seen in similar studies (*Gao, Liao & Li, 2008*; *Li et al., 2010*; *Zheng et al., 2011*; *Cheng & Zhang, 2010*). Taiwan has the largest numbers of total journal impact factor and total citation count among the three regions, indicating that it is the most influential region in China. This could be due to Taiwan having the highest number of total articles, far exceeding those from the other two regions. However, Hong Kong ranks the first among the three regions when mean journal impact factor and mean citation count are used. It may suggest that Hong Kong published higher-quality nursing research than Mainland China and Taiwan did. Moreover, Mainland China has the lowest mean journal impact factor and mean citation count. These findings indicate that, despite the rapid increase in the number of publications from Mainland China, articles from this region may be of lower quality and require improvement, compared to those from Taiwan and Hong Kong. Nevertheless, it should be recognized that the optimal tool for assessing the quality of publications remains controversial. The impact factor is intended to assess the quality of a journal, but not the quality of an individual article published in the journal (*Li et al., 2010*; *Grzybowski, 2009*). In general, the impact factors of the journals in which the articles were published suggested similar academic levels (*Lin et al., 2011*). Additionally, citation count may not be evenly distributed amongst articles in a journal, as a small number of articles would probably attract the bulk of citations (*Weale, Bailey & Lear, 2004*; *Seglen, 1997*). Moreover, the citations for an article are highly related to the quality of the article and the novelty of the findings. However, the number of citations is influenced by many factors, such as journal impact factor, geographic origin of the authors, whether they are English speaking, and the gender of the authors (*Eyre-Walker, 2013*; *Paris et al., 1998*; *Opthof, 1997*; *Leimu & Koricheva, 2005*).

*Journal of Clinical Nursing* was the most popular nursing journal in all the three regions. This result indicates that *Journal of Clinical Nursing* is the most influential journal in China. *Journal of Clinical Nursing*, *Journal of Advanced Nursing*, *International Journal of Nursing Studies*, and *Cancer Nursing* appeared in all top five journals listings in the three regions. It suggests that these journals may play an important role in promoting nursing research from China in knowledge-sharing efforts.

This study has some limitations. First, the data of another region in China, Macau, was collected, but the present study did not analyze it due to the low number of articles originating from this region. Second, the Web of Science database was used to identify nursing publications. Articles published in journals not cited in Web of Science were not included, although they might contribute to quantifying scientific production. Third, there is still no consensus on the indicators for assessing the quality of articles. The use of journal impact factor and citation count might not have been optimal in assessing some of the included articles. Fourth, although each included article was published in nursing journals, some articles may have other topics as their focus. Fifth, the journals were collected from the nursing category of the JCR, but multidisciplinary journals may also publish articles related to nursing, and such journals were not included in this study. Nevertheless, the 110 subspecialty nursing journals included in this survey represent the major international journals devoted to the discipline of nursing.

## CONCLUSION

There has been a significant increase in contributions to the field of nursing from China in the past decade, particularly from Mainland China. Taiwan is the most prolific publication region in China. Hong Kong had the highest-quality research outputs in terms of mean journal impact factor and mean citation count per article.

## ACKNOWLEDGEMENTS

We would like to thank our colleagues for their help on this study.

### Funding
The authors received no funding for this work.

### Competing Interests
The authors declare that they have no competing interests.

### Author Contributions
- Di Zhang conceived and designed the experiments, performed the experiments, analyzed the data, wrote the paper, reviewed drafts of the paper.
- Xiaming Wang performed the experiments, contributed reagents/materials/analysis tools, wrote the paper, reviewed drafts of the paper.
- Xueru Yuan analyzed the data, contributed reagents/materials/analysis tools, prepared figures and/or tables, reviewed drafts of the paper.

- Li Yang performed the experiments, wrote the paper, prepared figures and/or tables, reviewed drafts of the paper.
- Yu Xue conceived and designed the experiments, analyzed the data, reviewed drafts of the paper.
- Qian Xie conceived and designed the experiments, reviewed drafts of the paper.

**Data Deposition**

The raw data has been supplied as a Supplemental Dataset.

**Supplemental Information**

Supplemental information for this article can be found online at http://dx.doi.org/10.7717/peerj.1798#supplemental-information.

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
