# Peer review of "Scientific publications in nursing journals from Mainland China, Taiwan, and Hong Kong: a 10-year survey of the literature"

_PeerJ, doi:10.7717/peerj.1798_

## Round 0.1 · original submission · Major Revisions

Please pay special attention to the comments of Reviewer 3.

Reviewer 1 ·

Basic reporting

No Comments

Experimental design

No comments

Validity of the findings

See comments below.

Additional comments

This study is straightforward in answering a simple question of the contributions of Chinese authors to nursing research during a specific time period. The authors operationized the contribution as quality and quantity of publications in nursing journals by Chinese authors. Comparisons between different regions of China were also made. The study was descriptive in nature; no flaws in the conceptualization of major concepts, design and data collection process are noticed. The design is precise and conclusions are consistent with the results. Tables are straightforward as well.
Some redundancies in reporting are noticed. Typos or grammar problems are noticed in line 37, 88, 102, 142, and 164.
I would also revise the statement "… regression analysis was used to determine significant changes in time trend between 2005 and 2014" in line 73-74. The statement is problematic as regression is used to determine relationships, not changes.
In Line 179, the authors said “this study is the first survey…” I wouldn’t call this study a survey, which connotes a different type of research.

Reviewer 2 ·

Basic reporting

See below

Experimental design

See below

Validity of the findings

See below

Additional comments

My major concern is the political sensitive content. The authors should define who are the Chinese authors in the paper. I read and got the feel is the authors original from mainland China, Hong Kong, and Taiwan. But Macau should be included if the authors put mainland China, Hong Kong, Taiwan together. So the authors either only analysis the papers written by authors from mainland China or have all the four parts of China. So the paper need to do major revision if published

Reviewer 3 ·

Basic reporting

• The authors have not mentioned if they have received any fund for their study.
• I believe the English language of the manuscript should be improved. There are several grammatical and also technical mistakes throughout the text. I will mention some of them as examples. As an example: the first sentence of the abstract (line number 4) needs another verb tense as it is pointing a continuous event which has started in the past: “China has had/experienced/witnessed/achieved a remarkable progress…”. Similarly, in line 37 a present tense verb has been used for a process that has begun in the past: “In recent year, Chinese nursing has been rapid…” This grammar mistake is very common throughout the manuscript.
Line 38: putting commas between groups of 3 zeros makes it easier to read: 2,000,000
Line 39: “an” important instead of “a important”. In the same sentence I am not sure about the use of “force” in this context. Maybe “hub” can be an alternative?
Line 82: I guess one word is missing after “3.47-fold”. For instance, it could be “a 3.47-fold increase”?
Lines 109 and 110: “with 74 articles in Mainland China, 392 articles in Taiwan, and 146 articles in Hong Kong”, I guess the preposition “from” would fit better than “in”.
Line 134: instead of “have exceed” they might can say “has exceeded”?
Line 142: contribution “to” scientific …
Line 148: “relatively”, “needs”
Line 155: Instead of “these findings maybe because” I would suggest saying “this is probably because”
Line 160: “lower” instead of “low”
Line 161: “comparing to” instead of “comparing with”
Line 164: I did not understand what the authors meant by “the most influence journal”
Line 180: “there has been” instead of “there has”

I did not mention all mistakes. The text needs proofreading.
• Regarding the clarity of the language I would suggest the authors to replace “Chinese authors” with another term that represents exactly what they mean. When reading “Chinese authors” some may think the authors are talking about all Chinese authors, either living in China or not, whereas what they mean is authors who are based in Mainland China, Hong Kong and Taiwan. This is better to be explained more clearly in the text.

• The introduction needs significant improvement. It is written in two paragraphs while they are both giving a similar message. For instance, I do not see a big difference between the second sentence of the 1st paragraph and the second sentence of the 2nd paragraph. Or the last sentence of the 1st paragraph is very similar to the sentence in lines 45 and 46. The thirds sentence of the abstract saying “Chinese nursing is catching more and more attention”, I believe it is better that authors avoid such general sentences and try to replace them with more facts and evidences. The aim of study in the last sentence of the introduction is mentioned to be investigating the contribution of articles from China to the field of nursing. This aim is not in line with the methods and consequent results of the study. If the authors were investigating the contribution to the field, their results should have shown that for instance a certain percentage of the global research publications in nursing are coming from China, etc.
• In terms of the structure of the paper: The first page does not follow the journal template. Affiliations and Corresponding Author are not mentioned after the authors’ names. The headings in the abstract are not in line with the journal template. The authors have written “Objective” instead of “Background” and have used the heading “Conclusions” instead of “Discussion” in the abstract.
• I think the figures and the tables of the paper need a more comprehensive label, a label that includes the research field and the used database.

Experimental design

• The importance of this study has not been explained properly. It is not explained how this study can contribute to the knowledge. Also, there is no comparison with the increase in the global research output in the field. The reader should be convinced why this study is important. Why investigating the research output of China matters? Is it different from the global trend? Is it only because China has a high population of registered nurses? The arguments are not strong enough. It is also not mentioned anywhere why the authors have chosen this 10-year period.
• In terms of the name of the used database in this paper I guess the authors have used the Web of Science Core Collection. I am guessing so as they have selected the journal titles from Journal Citation Report which only includes the journals indexed in the core collection of the Web of Science not its “all databases”. In this case, the name of the used database should be changed to the “Web of Science Core Collection”.

• In describing the “search strategy” in Materials & Methods section of the paper, lines number 52 and 53, the authors have mentioned four previous studies which have been used for designing their research. These four articles do not share the same methodology as the database used in them is not the same. The first two papers have retrieved the articles from pubmed, while the other two have used Web of Science. In the methods section I would expect more details about the exact search strategy of this study in a way that makes it replicable. For instance, after reading the methods and materials section, it is still not clear to me how the authors have reached the citation reports.

• The basic methodology of the study sounds correct to me, but my question is why the authors have not used the “Research Areas” available on the refine panel of the Web of Science Core Collection for retrieving the articles. At the end, by searching the articles published in journals found on Journal Citation Reports, the authors have only accessed the publications in the Core Collection, not in all databases. Therefore, they could have used the Research Areas refining option on Web of Science Core Collection. Research Areas refining option refines the search results based on the scope of the journals where the articles have been published. It is an easier way than finding all the journal titles from the Journal Citation Report, then searching the publications within each journal, one by one. I believe both methods have the same limitation that there may be papers in research field of nursing that have been published in non-nursing journals, so they will not come up in the search results. The positive point of my suggested method over the method used in this study is that it also may capture some nursing papers which have been published in multidisciplinary journals such as “Science”. Sometimes, but not always, the editors in Thomson Reuters would assign the research areas of individual papers in these high impact factor multidisciplinary journals manually. Consequently, those papers would be linked to, for instance, nursing research area even though not being published in a nursing journal. However, in both methods publications in general medical journals such as Lancet will be missed. (The authors have mentioned this limitation at the end of their discussion).
• The other limitation of both methods is that the articles with non-nursing topics which have been published in these journals will be included in the dataset. For instance, one of the journals in the dataset of the current study is “Perspectives in Psychiatric Care”. Its categories have been identified as both “nursing” and “psychiatry”. Even if psychiatric articles, with no relevance to nursing, are published in that journal would be included in the dataset. But my question is whether the authors of the paper have excluded these irrelevant articles from their dataset or not. I see that the authors have the full list of all articles. If they have excluded irrelevant articles, this should be mentioned in the paper.
• Another point is that in the methods the authors have mentioned that “Two reviewers independently conducted study selection and data extraction. Disagreements were resolved by discussion, and a third reviewer was consulted when necessary”. Did the researchers only plan to follow this method, or were there any controversial cases at the end which required a third person comment? If there was such a need, it is better to mention it in the discussion, if not, it is better to omit this part from the methods.
• Impact Factor is an indicator which is basically defined for journals, not individual articles. Therefore, the notion of “total impact factor of published papers” requires more clarification in the methods section.
• Regarding table 1, I am not very interested in knowing this data as we already know that there is a significant difference between the number of publications of these three regions, so it is not very surprising seeing that the summed up amount of the impact factors of the journals published those articles, or similarly the total number of citations received by all those publications is significantly different. What matters is the average numbers. The authors have also mentioned this in the discussion. What I mean is that I find this table unnecessary.
• Looking at the raw data of the impact factors of the journals I can see that there should not be outliers, but still I would suggest the authors to at least mention the range of journal impact factors for each region. For instance, the impact factor of the journals published the articles from Taiwan was between X and Y.
• The way the results on High-Impact Nursing Journals is described (line 100 and table 5) the reader may think the authors are talking about the journals from China (Chinese journals), while they are talking about the articles by authors based in China which are published in those journals. This section needs a more clear description. Another question is if the authors have thought of a way to capture the publications in high impact general medicine/multidisciplinary journals such as Lancet/NEJM/Science, etc.
• Line 107, the “Popular nursing journals” part in the results section also needs more clarification. I would suggest the authors to describe their definition of “popular”.

Validity of the findings

• Discussion: Lines 108 and 109: I guess China has always been contributing to the science throughout the history. It is more correct to say that previous studies have reported the “increase” in China’s contribution to the science. In any case, I do not find the lines 117 to 122 adding anything new to the paper. These lines can be omitted.
• It is good that the authors have read the relevant studies from China, but it would have been more helpful to see some comparison with the global trends in research output in nursing, or at least comparing with few other countries in the discussion. The authors must elaborate more on their arguments in the discussion. Only citing other articles is not enough.
• In lines 136-139 of the discussion the authors have poorly tried to mention the possible reasons behind the increase in China’s nursing research output. They have to elaborate on this. Also, lines 140-142 need a reference.
• Line 150: “medical reform”? Please leave a reference for the reader.
• One of the aims of this study was assessing the quality of publications and the number of citations and the impact factor of journals publishing the articles are used as the proxy of the quality of scientific papers. In lines 151 and 152 the authors mention the controversy in choosing an indicator for measuring quality, but this needs further elaboration on the controversy and challenges. It needs citation to papers that have discussed these challenges.
• Other limitations of the study: 1) the use of journal impact factor and citation count for assessment of the quality of articles 2) the possibility of inclusion of articles in the dataset that despite being published in nursing journals may be on other topics 3) not capturing the articles published in multidisciplinary journals such as Science
• I would suggest the authors to use the correct full terms while talking about indicators, such as “journal impact factor” or “citation count”/”the number of citations
• The first sentence of conclusion is not a conclusion!

Additional comments

no comments.

---

## Round 0.2 · accepted · Accept

Thank you for the revised manuscript.

Reviewer 2 ·

Basic reporting

No comments

Experimental design

No comments

Validity of the findings

No comments

Additional comments

Well done.